# Food Parenting Practices and Feeding Styles and Their Relations with Weight Status in Children in Latin America and the Caribbean

**DOI:** 10.3390/ijerph19042027

**Published:** 2022-02-11

**Authors:** Luisa Pérez, Marcela Vizcarra, Sheryl O. Hughes, Maria A. Papaioannou

**Affiliations:** 1Carrera de Nutrición y Dietética, Facultad de Medicina—Clínica Alemana, Universidad del Desarrollo, Av. Plaza 680, Santiago 7610658, Chile; le.perez@udd.cl; 2Food Behavior Research Center, School of Nutrition and Dietetics, College of Pharmacy, University of Valparaíso, Av. Gran Bretaña 1093, Valparaíso 2360102, Chile; 3USDA/ARS Children’s Nutrition Research Center, Baylor College of Medicine, 1100 Bates Ave, Houston, TX 77030, USA; shughes@bcm.edu (S.O.H.); papaioan@bcm.edu (M.A.P.)

**Keywords:** food parenting practices, feeding practices, feeding styles, Latin American and Caribbean, child weight status

## Abstract

While a growing body of literature looks at the associations between food parenting practices, and feeding styles, and child’s weight status in developed countries, little is known for less developed countries, in general, and the Latin America and the Caribbean (LAC) region, in particular. This study systemically reviews and synthesizes existing evidence on the associations between child caregivers’ food parenting practices and feeding styles and 2 to 12-year-old child weight status. Keywords were used to search in PubMed, Web of Science, PsycINFO, and Cumulative Index to Nursing and Allied Health Literature. Among the ten eligible articles, all of them reported significant associations between food parenting practices and feeding styles and child weight status. Existing studies have limitations, mainly related to cross-sectional convenience samples, which limit the generalization of the results. Additionally, small sample, heterogeneous feeding measures and weight related outcomes were other limitations. Future research is needed to understand caregiver–child interactions in the food situation and its link to child weight status in 2 to 12-year-old children in areas of LAC with diverse forms of malnutrition and contextual factors of countries.

## 1. Introduction

One of the main international commitments for sustainable development in Latin America and the Caribbean region (LAC) is to reach the dual goals of reducing all forms of malnutrition (two indicators of it are the prevalence of stunting and overweight in children) [1]. LAC has displayed a reduction in stunting in children from 22.7% in 1990 to 9% in 2019 [1], from 12.8 to 4.7 million children, while the prevalence of overweight in children changed from 6.2% in 1990 to 7.5% in 2019 affecting 3.9 millions of children [1]. Diversity of realities regarding the nutrition situation between and within countries are reflected in their nutritional indicators. Some countries are affected by different levels and types of double burden of malnutrition at the household level (i.e., mother’s overweight: Body Mass Index (BMI) ≥ 25 m/kg^2^; and child under five years with stunting: Height-for-age z score < 2) such as in Ecuador (25.2%), Guatemala (48%) and Uruguay (10.7%), whereas in other countries like Chile, overweight is the main public health concern (9.3%) and stunting (1.8%) is considered eradicated [2].

Malnutrition in LAC is frequent. It presents in lagging territories due to stunting or overweight (a lagging territory is defined as one with a prevalence of stunting/overweight higher than the respective averages at the national level) [1]. Among countries with available information, 23 have lagging territories due to stunting in children under 5 [1]. Additionally, there is a vast difference of malnutrition among lagging territories due to overweight in children under 5. Although overweight has constantly been increasing in the region, some territories tend to concentrate in countries including Brazil (southwest), Chile (south), Mexico (north), and Argentina (south) [1]. Therefore, the distribution of malnutrition (overweight and stunting) is not homogenous within and among countries, making it necessary to address all forms of malnutrition in LAC to promote children’s maximum potential and focus on different influences, including the family environment, where food parenting practices and feeding styles play a role.

One reason for this double burden of malnutrition and rising problem of overweight in LAC is the change in patterns of food consumption brought on by changes in their food systems [3]. For example, people consume food with low nutritional quality due to the low availability of nutrient-rich foods as well as the increased availability of ultra-processed foods [3]. Although there have been international efforts aimed toward ending food insecurity, hunger, and malnutrition, the burden of both ends of malnutrition, under- and over-nutrition, are concerning and attention to both nutritional issues and its influences is required in LAC [4].

These variations in nutritional situations among countries of LAC have occurred as a result of the Iberian conquering and later colonialism, which have differentially impacted the economy, demography, and cultural traditions (e.g., development of similar traits such as language) in countries of LAC [2,5]. Latin America consists of 39 countries where 652 million people live [6]. These countries are located in South America, Mexico, Central America, and the islands of the Caribbean, where their inhabitants were mainly colonized by Spain and Portugal [7]. South American countries include Venezuela, Colombia, Ecuador, Peru, Brazil, Chile, Argentina, Uruguay, and Bolivia. Central American countries include Panama, Costa Rica, Nicaragua, Honduras, El Salvador, Guatemala, and Belize [7]. In the Caribbean, the islands that were colonized by Spain were Cuba and the Dominican Republic. Besides the similar cultural traits in LAC countries, defining any cultural practices and habits (such as feeding and food intaking) in Latin America is as complex as understanding the continental experience as an integrated community [8]. In fact, looking for certain kinds of practices, public policies, or living experiences in this specific landscape implies to stretch geographical and cultural frontiers that are far from homogeneous. For instance, such is the case of cross-borders zones, like the tri-border area between Paraguay, Argentina, and Brazil, as well as Northern Chile and South Peru [8]. All of these international zones eventually share more people, goods, and identities than actual cities connected to their national metropolis [8].

The socio-cultural and territorial context conveyed at the home food environment influences children to appropriately develop eating habits and have a healthy growth [9]. At home, parenting practices involve the behaviors or actions that parents apply in an effort to guide the attitudes, behaviors, or beliefs of their children [10]. These practices differ according to the situations in which parents and children interact, such as feeding [10]. Thus, food parenting practices, are defined as goal- or content-specific parent behaviors to feed their child [11]. Given the large number of these practices, Vaughn et al. [10] offers higher order categories of food parenting practices. These categories involve: (1) coercive control, (2) structure and (3) autonomy support. Practices within the category of coercive control refer to parent-centered feeding strategies that can negatively impact children’s eating behaviors and preferences [10]. Examples of these food parenting practices are food restriction, pressure to eat, threats and bribes, and the use of foods to control negative emotions in the child [12]. These practices are considered not appropriate because they can disrupt the ability of the child to self-regulate their eating [12]. Food parenting practices in the category of structure comprise a wide diversity of strategies that enable the child to become more competent through the organization of the environment where s/he eats. Examples of food parenting practices in this category include healthy/limited guided choices, monitoring, food related meal and snack routines, and modeling. The autonomy support category includes food parenting practices that provide opportunities and support for the child to become autonomous in choosing their foods while considering his/her developmental stage [10]. Examples of food parenting practices in this category are praising the child to eat, reasoning when communicating with the child to promote understanding about limits, and rules related to foods, while also creating an emotionally supportive environment [10].

Different to food parenting practices, feeding styles refer to the overall attitude of parents that results in general patterns of behaviors that parents apply when feeding their children based on the dimensions of responsiveness and demandingness [13]. Demandingness refers to how much parents encourage child eating (that is, how demanding they are during the eating experience) and responsiveness refers to how parents encourage eating (that is, the level of nurturance parents use in directing child eating) [13]. Caregivers are categorized into feeding styles based on high and low dimensions of responsiveness and demandingness as follows: (1) authoritative style (high demandingness, high responsiveness) is characterized by the use of non-directive behaviors while still setting appropriate boundaries; (2) authoritarian style (high demandingness, low responsiveness) is characterized by strict demands on the child, but without considering the child’s individual needs; (3) indulgent style (low demandingness, high responsiveness) is characterized by few demands and boundaries, but the few demands made are supportive; and (4) uninvolved style (low demandingness, low responsiveness) is characterized by little involvement and support around feeding.

Food parenting practices and feeding styles are recognized as influences on child weight status in diverse populations mostly living in developed countries [14]. For example, indulgent and uninvolved feeding styles have been associated with higher weight status and unhealthy eating [15]. Additionally, restriction has been associated with higher and lower BMI in children [14]. Given the ways that parents or other caregivers feed their children may be affected by geopolitical and socio-cultural factors [9], similarities and differences in LAC may involve different associations between food parenting practices and feeding styles, and child weight status. Cultural aspects are reflected at the micro-level of family since traditions and customs involve food as a vital aspect of everyday life [16] Parents’ beliefs and attitudes in relation to mealtimes and how children should be fed are examples of cultural aspects that may influence how parents interact with their children in food-related contexts [17]. Thus, there is a need to review existing literature on parents’ feeding behaviors and their relation to child weight in other parts of the world. Countries in LAC, specifically, are facing unequal types of malnutrition and environmental changes, which combined with socio-cultural factors may influence parents’ feeding and their relation to child weight, thus findings from developed countries cannot be extrapolated to countries with different contexts [18].

In this context, this systematic review aims to explore food parenting practices and feeding styles in LAC and its objectives are: (1) to summarize existing evidence of the relations between food parenting practices and feeding styles and the weight status of preschool or school age children residing in LAC countries; (2) to describe findings and limitations of the studies conducted in LAC countries and (3) to suggest further research in the region of LAC. This review focuses on 2 to 12-year-old children because this is the age range they develop eating behaviors, and while they are still dependent on their caregivers, they are becoming more independent from them [19]. Food parenting practices and feeding styles may have long lasting effects on the eating behaviors, habits and growth of children of this age range [20,21].

## 2. Materials and Methods

### 2.1. Study Eligibility Criteria

Studies that met all the following criteria were included in the review: study design (longitudinal, case control, randomized control trial, cross-sectional study); population (caregivers and their children between the ages of 2 and 12); country (countries in Latin America or the Caribbean that shared Spanish or Portuguese language because their cultural similarities from the Iberian background of colonization); main outcome (child weight status referring to BMI, BMI z-score or child weight status category); article type (peer-reviewed publication); and language (English, Spanish, and Portuguese). The exclusion criteria were: children with eating disorders; review paper; case study; qualitative article; and theoretical article.

### 2.2. Search Strategy

The systematic review protocol described in the Preferred Reporting Items for Systemic Reviews and Meta-Analysis statement [22] was adopted to guide the review process. Keyword searches were performed in PubMed, Web of science, PsycINFO, and Cumulative Index to Nursing and Allied Health Literature (CINAHL) on 4th February 2021. There were no restrictions regarding date of publication. The search algorithms for each database were adjusted to optimize the literature search; the PubMed search was used as the basis for the search in the other databases. The search algorithm for PubMed included all possible combinations of keywords from the following four groups: (1) (“Feeding behavior” [Mesh] OR “maternal feeding behavior” OR “parent feeding” OR “caregiver feeding” OR “responsive feeding” OR “non-responsive feeding” OR “food parenting practices” OR “parent food practices” OR “Food practices” OR “feeding style*” OR “feeding practices” OR “feeding strategy”) AND (child [Mesh] OR preschool [Mesh]) AND (body mass index [Mesh]OR body weight [Mesh] OR weight status OR overweight OR obesity, OR BMI OR “BMI z score” OR underweight OR low weight) AND (Latin America [Mesh] OR South America [Mesh] OR Central America [Mesh] OR Latin America* OR latino OR Caribbean OR Chile* OR Argentin* OR Brazil* OR Peru* OR Bolivia* OR Uruguay* OR Colombia* OR Venezuela* OR Mexic* OR Ecuador* OR Guatemal* OR Cuba* OR Honduras OR Honduran OR Paraguay* OR el Salvador OR Salvadorean OR Nicaragua* OR Costa Rica* OR Dominican Republic* OR Panama*). Titles and abstracts of the articles identified through keyword search and MESH terms were screened against the study selection criteria. Potentially relevant articles were retrieved for a full-text-evaluation. Two authors (LP and MV) of this review independently conducted the title and abstract screening, as well as the retrieval, with disagreement resolved through discussion between the two researchers.

### 2.3. Data Extraction

Using a standardized data extraction form, the two authors independently read and extracted data for all included articles to gather the following: (1) author(s)/publication year/country; (2) sample’s characteristics and study design; (3) study aim(s); (4) measure of food parenting practices or feeding styles; (5) measures of child weight status; and (6) main findings. The authors discussed and resolved discrepancies found. We present results of our review as a narrative summary.

### 2.4. Study Quality Assessment

Study quality was assessed based on an adjusted version of the Strengthening in the Reporting of Observational Studies in Epidemiology statement (STROBE) proposed by Lindsay et al. [23]. The proposed criteria were as follows: (1) Is the study longitudinal? (2) Does the paper describe the participants’ eligibility criteria? (3) Were study participants randomly selected (or representative of the study population)? (4) Did the paper report information about the measures (including validity in previous studies), including references used to assess food parenting practices? (5) Did the study include information about acceptable reliability of the instrument used to assess food parenting practices? (6) Did the paper report how child weight status was assessed in children participating in the study? (7) Did the study provide information about power calculation to detect hypothesized relationships? (8) Did the study report the number of individuals who completed each of the different measures? (9) Did the participants/respondents complete at least 80% of measures? (10) Did analyses account for confounding factors? 

Two authors (LP and MV) independently scored each study based on these 0–10 criteria, with disagreement resolved through discussion. The Cohen’s Kappa was 0.61, with an 81% agreement between the two authors. Scores for each criterion ranged from 0 to 1, depending on whether the criterion was unmentioned or unmet (0) or met (1). The possible total study score ranged between 0 to 10. Study quality score helped measure the strength of study evidence, but was not used to determine inclusion of studies. A study was considered good-quality when a score of 7 or higher was obtained from the 0–10 checklist criteria [23].

## 3. Results

### 3.1. Study Selection

As Figure 1 shows, the search strategy generated 1794 unduplicated articles identified through keyword and reference search, 1775 of them were excluded during title and abstract screening. The remaining 19 articles were reviewed in their entirety, and 9 of them were excluded for not meeting the study selection criteria: studied populations did not live in Latin America or the Caribbean [13,24,25,26]; in addition to not being conducted in LAC, one of these studies used a qualitative method [25]; weight status was not included as an outcome variable [27,28,29,30], and one of them also studied children who were out of the age range of 2 to 12 years old [31]. The remaining 10 articles [32,33,34,35,36,37,38,39,40,41] were included in the review.

### 3.2. Summary of Study Characteristics

Table 1 reports a summary of the 10 articles included in the review. Five studies were conducted in Brazil, two in Chile and three in Mexico, while no studies were found in the Caribbean. This small number of studies from only three countries limits the findings of this review. Most participants were recruited from urban areas, mainly from schools. Sample sizes ranged from 91 to 1071 pairs of children and their caregivers, with mothers as the main caregiver across studies. Children’s ages ranged between 2 and 11.98 years old. The three studies that reported child gender had a balanced distribution between boys and girls, ranging from 49.08% to 53.9% for girls.

Regarding data extraction (see Appendix A), participants across all studies were recruited from schools and day care centers or kindergartens [32,33,34,35,36,37,38,39,40,41]. The parent’s education level was insufficiently reported in the studies. Six studies reported parent’s education [33,34,35,38,40,41]. Most of the studies reported medium or high level of education level [33,34,35,38,40,41], and seven studies reported caregiver’s age [33,34,35,38,39,40,41]. Among the selected studies, eight employed a cross-sectional design [32,33,34,35,36,38,39,41], one used a longitudinal design [40], and one used a case control design [37]. We included this observational study design because food parenting practices was considered as a risk factor and the odds ratio of the child to be obese was the outcome. Although case control studies are different from the other observational studies included in this review, we considered it met the inclusion criteria of using food parenting practices as an independent variable and weight status as outcome variable.

Some studies included more than one measure of food parenting practices (e.g., CFQ and CFSQ). Four studies used the Child Feeding Questionnaire (CFQ) [32,36,39,40]. One study used the Caregiver’s Feeding Style Questionnaire (CFSQ) [34] and one used the Toddler Feeding Questionnaire [40]. Three studies used the Comprehensive Feeding Practices Questionnaire (CFPQ) [33,38,41]. Finally, one study used a questionnaire developed by the researchers, the only one not previously validated [37]. The relations between food parenting practices and child weight status included measures of BMI (Body Mass Index), BMI-z score or risk of overweight/obesity.

#### 3.2.1. Effect of Food Parenting Practices and Feeding Styles on Children’s Weight Outcomes

Nine out of ten studies included in this review found a significant relationship between food parenting practices or feeding styles with child weight status. We present the associations between food parenting practices with child weight status according to the categorization of Vaughn et al. [10]: coercive control, structure, and autonomy support (Figure 2).

#### 3.2.2. Feeding Practices: Coercive, Structure and Autonomy Support

Coercive feeding practices. Feeding practices categorized in this group were restriction, pressure to eat, and the use of food as a reward. Regarding restriction, seven out of 10 studies assessed a relation between this feeding practice and weight status in the child, including definitions of restriction for weight control and health. Four observational cross-sectional studies [32,38,39,41] indicated that restriction was related to higher child BMI z-scores. One case–control study reported higher likelihood of the child of being overweight [37]. One study did not find any association between restriction and child weight status [36], and the only longitudinal study reported that only food restriction at baseline predicted lower child BMI z-scores in girls [40]. These findings are presented in Table 2.

Six of the seven studies that measured pressure to eat found a relation with child weight status. Six cross-sectional studies found a relation with child weight status [32,33,35,36,39,40,41], of which five studies found a negative cross-sectional association between pressure to eat and child weight outcomes [32,33,36,39,41]. One study out of the six cross-sectional studies indicated that higher child weight was associated with less likelihood of parents to apply pressure [33]; while another study reported that normal weight in boys was associated with higher parent pressure [36]. In one study, pressure to eat was associated with lower likelihood of the child being overweight [32], and parents applied more pressure over obese children in another study in Mexico [35]. One longitudinal study did not find a relation between this food parenting practice and child weight status [40].

Of the two studies that assessed the relation between the use of food as a reward and child weight status, one cross-sectional study did not find any significant association [41]. One longitudinal study found that parents using food as a reward was associated with higher BMI z-scores [40].

Structural feeding practices. Structural feeding practices measured in the studies included in this review were monitoring [32,35,36,39,40,41], limit setting [35], healthy eating guided choices [41] and having a schedule to eat, and eating in family [37]. None of the six studies examining the relation between the monitoring and child weight status found significant associations [32,35,36,39,40,41]. Healthy eating guiding choices and limit setting were associated with lower [41] and higher BMI z-score in children [41], respectively. Having a schedule to eat and eating in family were not associated with child weight status [37].

Autonomy support. Reinforcement defined as by praising the child to eat, was the only food parenting practice that was explored in this category, which was used more in children with underweight [35].

Feeding styles. Feeding styles were measured in two studies. One cross-sectional study found that children whose parents had the uninvolved and the indulgent feeding styles were those that had the highest mean BMI [34]. In the longitudinal study, high child BMI z-score was associated with higher indulgent parent feeding after a one year follow-up in boys, while indulgent feeding was associated with higher values of child BMI z-score after a year in both boys and girls [40].

Some of the selected studies reported results of food parenting practices and feeding styles, and children’s weight status according to child age. We present the main findings by age group: 2 to 5 years old and 6 to 12 years old.

Age group 2 to 5 years. Among food parenting practices, healthy eating guidance (β = −0.36, *p* < 0.05) and pressure to eat (β = −0.22, *p* < 0.05) were negatively associated with child BMI z-score [41]. Another study found a positive association between children’s BMI-z scores and restriction in boys (rho = 0.19, *p* < 0.001) and in girls (rho = 0.27 *p* < 0.001), and a negative association between pressure to eat in boys (rho = −0.30 in boys, *p* < 0.001) and girls (rho = −0.36, *p* < 0.001) and BMI z-scores [39].

In terms of feeding styles, a longitudinal study reported that higher indulgent feeding style was positively associated with child BMI z-scores in the total sample after a year follow-up (β = 0.36, *p* = 0.001), especially in boys (β = 0.39, *p* = 0.008) [40]. In girls, higher use of food as a reward at baseline positively associated with higher BMI z-scores after 1 year follow-up (β = 0.31; *p* = 0.04), while food restriction at baseline predicted lower BMI z-scores at follow-up (β = −0.34; *p* = 0.03) [40].

Age group 6 to 12 years. Regarding food parenting practices, restriction was consistently associated with higher child weight. Restriction for health was positively associated with child BMI z-scores (β = 0.09, *p* < 0.05) [41]; and higher food restriction (mean = 25.38, SD = 5.09, *p* = 0.001) and monitoring (mean = 7.08, SD = 2.88, *p* = 0.007) was exerted by parents whose children were overweight compared to those parents whose children had a normal weight status [32]. Parents’ constant restriction on the amount of foods the children use to consume is associated with a higher risk for obesity in the child (OR = 62.9, IC: 5.37–92.08, *p* < 0.0012) [37].

Pressure to eat was negatively correlated with BMI z-score in cross-sectional analyses only in boys (r = −0.21, *p* < 0.05) [36], while in retrospective analysis, restriction, pressure to eat, and monitoring did not account for any variance in child BMI z-scores [36]. In another study, pressure to eat was related to lower odds of being overweight in the child (OR = 0.70, *p* = 0.01) [32].

Related to feeding styles, parents who were uninvolved had children with the highest BMI (mean = 16.84, SD = 2.7) followed by the indulgent feeding style (mean = 16.26, SD = 3.9) [34]. The researchers did not report differences of child BMI between feeding styles.

#### 3.2.3. Study Quality Assessment

The results of the study quality assessment are shown in Table 3. Overall, only two studies were categorized with a good quality according to the criteria. The 10 studies included in this review scored from 4 to 8 in the quality assessment. Eight studies described the validity of feeding measures [33,34,35,36,38,39,40,41]. Two studies randomly selected their sample [35,36]. Seven studies reported the reliability of the feeding measures used [32,33,34,36,40,41]. One of the studies was longitudinal [40].

## 4. Discussion

The main objective of this review was to summarize existing evidence of the associations between parental feeding and weight status in 2 to 12-year-old children in LAC, countries located in Central America, South America, and Caribbean islands, where their inhabitants were colonized by Spain or Portugal. Only ten studies met the criteria and examined the association between food parenting practices or feeding styles and child weight status. Overall, food parenting practices were more frequently measured in relation to child weight status than feeding styles in cross-sectional studies.

Among coercive feeding practices, restriction was associated with higher child BMI (β = 0.09, *p* < 0.05 in restriction for health in school-age children; β = 0.28, *p* < 0.05 in restriction for weight control in preschool- and school-age children [41]) and higher risk for overweight/obesity (OR = 62.69; *p* = 0.0012 [37]; OR = 2.18, *p* < 0.001 [38]; OR = 1.36, *p* < 0.001) [32]; in boys OR = 4.1, *p* = 0.001 [39]) in children in cross-sectional studies. Pressure to eat was associated with lower child BMI (β = −0.22, *p* < 0.05 in preschoolers [41]) and normal weight/lower risk of the child having overweight cross-sectionally (OR = 0.86, *p* = 0.008 [33]; OR = 0.709, *p* = 0.010 [32]). The use of food as reward was related to higher child weight status only in one longitudinal study, limiting findings of this review regarding this food parenting practice (β = 0.31; *p* = 0.04) [40]. Food parenting practices that promote structure in the child during eating occasions included mainly monitoring the child to eat, but generally it did not relate to child weight status, except in one study [32], which found that parents monitored more the diet of children with overweight. Only one study assessed healthy eating guidance, which was related to lower BMI [41], while another study found no significant relation between child weight status and limit setting, having a schedule for meals, and eating together as a family [37]. Complementing the child to eat, named as reinforcement in the study by Flores-Peña et al. [35], was the only food parenting practice in the category of autonomy support and no significant association with child weight status was found. All 10 studies discussed diverse influences involved in the associations between food parenting practices and child weight status [32,33,34,35,36,37,38,39,40,41]. These influences were attitudes, perceptions and concerns of child weight status, the role of parent weight status, socio-demographic characteristics, and behavioral and child characteristics that influence parent feeding. Thus, examining the motivations behind the feeding behaviors as well as the characteristics of parents and their children can enrich what is already known by highlighting the nuances and complexities in the mechanisms that explain how feeding practices and child weight status relate.

Interestingly, the association between restriction and higher child BMI and the likelihood of the child being overweight was found in this review regardless of the variety of definitions for this food parenting practice (restriction for weight control, restriction for health, etc.). These associations were similar to another review of studies that reported studies conducted mainly in the United States, Europe, and Australia [14]. The relation between restriction and weight status in children aged 4 to 12 years was stronger in most of the studies with a cross-sectional design, but the review reported a weak or no association in prospective studies [14]. In the only longitudinal study in LAC, food restriction predicted lower BMI in a small sample of preschoolers. These findings were similar to a longitudinal study in Australia, where restriction was associated with lower BMI-z scores after three years only in young children. In another longitudinal study following 8 to 10 year old Mexican American children for two years, higher child BMI-z scores at baseline predicted restriction; however the reverse prediction was not supported [42]. Therefore, evidence suggests that restriction may be a consequence of parental awareness of child overweight, but not the reverse, while restriction of energy rich foods may have a protective effect on young children, who are more dependent on caregivers. Other studies have also described a distinction between overt and covert restriction, adding an extra complexity to research this food parenting practice in LAC when parents restrict children in manners the child realizes about it [43].

Pressure to eat was associated with lower risk of children being overweight/obese and lower child BMI in our review. Two other reviews measuring pressure to eat also found this trend [14,44]. One study conducted in Mexico (included in this review), found that parents pressure children with obesity to eat more. This dissimilar finding may be due to cultural motivations in food parenting practices of Mexican parents, who may pressure their child to eat regardless of child weight status because of the positive attitude toward chubby children [45].

Coercive food parenting practices are defined as those that reflect attempts to dominate, pressure, or impose the parents’ will upon the child [10] and are considered an external over-control in the food context. In this review, the use of more external excessive control on children are inversely related to child weight status: restriction relates to higher child weight, while pressure to eat relates to lower child weight in cross-sectional studies. It is interesting that coercive feeding distinctively relates to child weight status in this review, but very little has been studied regarding other practices that consider the child’s needs and encourage autonomy. Additionally, coercive feeding is considered a disruption in the child’s ability to self-regulate their eating and in the development of their autonomy for self-feeding [46]. However, it is still necessary to further understand the association between different conceptualizations of coercive feeding practices such as different constructs of restriction and child weight status in different countries of LAC, which have shown differential associations with child weight status.

One parent characteristic that has been studied in the LAC region limitedly is the parental feeding style. Indulgent feeding was associated with higher child BMI in a longitudinal study in Brazil, particularly in boys, whereas parents with an indulgent or uninvolved feeding style had children with higher BMI compared to the other feeding styles in one cross-sectional study in Mexico. Vollmer and Mobley [47] discussed that parent responsiveness, one of the dimensions of feeding styles, influences whether children are more or less receptive of the feeding practices that parents apply. Although more research on how feeding styles develop is needed and its conceptualization as a context for feeding practices [47,48], feeding styles may have an indirect influence on children’s weight status because children could be more or less receptive according to parent’s dimensions of feeding styles.

Consistent with this review, the indulgent feeding style is associated with higher child BMI in cross-sectional studies [49,50,51] and a longitudinal study in the US [20]. The use of the CFSQ for measuring the specific construct of feeding enables comparison across countries, but very little is known about which are the most prevalent feeding styles in LAC and how they are associated with child weight status. Feeding styles are relevant because they can moderate the relations between food parenting practices and child weight status and other weight-related outcomes [15,47]. It has been proposed that feeding styles relate to child weight indirectly [47]. For instance, children that eat emotionally have higher child weight only when their parents have an uninvolved feeding style. Another example is that children eat more fruits and vegetables when parents restrict junk foods only when parents have an indulgent feeding style [52]. Therefore, indulgent parents who restrict unhealthy foods improve their child’s diet, which indirectly can influence the child weight status.

### 4.1. Limitations of the Studies Conducted in LAC Countries

This review attempted to examine the links between food parenting practices, feeding styles and child weight status in LAC. Nevertheless, no studies examining these links have been conducted in the Caribbean, which limits the findings of this review to only Latin America and in only three countries (Brazil, Mexico, and Chile). By summarizing the existing research about these links, this review provides evidence of the growing body of research in the feeding domain related to weight outcomes, mainly conducted in developed countries.

The results should be considered with caution due to the study designs included in the review. The cross-sectional nature of most of the studies cannot inform the direction of the associations, or causality (only one study was longitudinal, but it involved a small sample). Additionally, the use of convenience samples and the small sample size of some studies confines the generalization of the findings.

Additionally, the feeding measures were based on self-report, which can involve a gap between actual food parenting practices and those that are reported given the social desirability related to self-report assessments. The diversity of questionnaires across the ten studies may have also contributed to varying results and whether these results can be compared.

Another limitation of the studies of this review is the absence of control for covariates linked to weight status in those studies assessing the associations between child weight status and parent feeding [47]. The aspects influencing child weight status or feeding include parental characteristics such as maternal education, maternal depression, and parent weight status, and child characteristics such as temperament, and appetite traits, among others [47,53]. Thus, the strength of the associations in models not controlling by covariates requires being analyzed with caution because including these covariates may change the strength or null associations.

Additionally, ethnicity is another aspect that could limit the findings of the relationships between food parenting practices and feeding styles and child weight status. Cultural particularities such as motivations, traditions and identity may carry out a variety of feeding practices [16,54]. For instance, Afro-Caribbean parents applied food restriction to control weight more frequently and had the lowest frequency of food monitoring, while British and German parents had similar frequencies of these two practices [9]. The associations with weight-related outcomes according to food parenting practices also varied. Only the group of Afro-Caribbean parents showed significant correlations between child BMI standard deviations and restriction to child weight. Therefore, ethnicity would have been a useful factor to examine whether there are specific associations between feeding styles and food parenting practices, and child weight status in diverse ethnic groups existing in LAC.

### 4.2. Recommendations in Further Research

The 10 studies included in this review were conducted in three countries experiencing high prevalence of overweight as a main public health concern: Brazil (7.3% in children under 5 years of age and 20.5% in children older than 5), Chile (9.3% in children under 5 years of age and 31% in children older than 5), and Mexico (9% in children under 5 years of age and 43.9% in children older than 5) [2]. Future research is needed to understand caregiver–child interactions in feeding situations and their link to child weight status in 2 to 12 years old children in areas of LAC with diverse forms of malnutrition and contextual factors of countries. Culture and context may impose similar and different effects in the way that parents interact with their children in the food context and nutrition situation in countries of the region. For example, in Peru, where stunting and micronutrient deficiency are nutritional issues, a lack of appetite in children under two years old led caregivers to encourage children to eat more [55]. Therefore, the associations found in this review (e.g., positive association between restriction and child weight), may not necessarily occur in families residing in countries where there is a double burden of malnutrition or there have been a long-lasting prevalence of children being stunted and wasted.

As the majority of the studies conducted in LAC are cross-sectional, future studies should include a longitudinal design to better understand the direction of the associations between feeding and child weight status. Finally, research design involving observations as a complement to self-report measures can improve the understanding of how parents interact with their children in the food context to limit biases related to social desirability.

In addition, little is known about a wide array of food parenting practices used in LAC. Some of the studies in this review used subscales of the CFQ, a widely used instrument to measure feeding, in particular restriction, monitoring, and pressure the child to eat, allowing comparison between studies as they use the same conceptualization of these food parenting practices. However, the conceptualization of certain food parenting practices to compare between studies in this review may vary. For instance, the CPFQ identifies restrictions applied with different goals by parents such as restrictions for weight or restrictions for health. Thus, studying diverse food parenting practices including same conceptualizations and measures of parent feeding is needed to allow comparison between studies of countries in LAC.

The CPFQ measures multiple food parenting practices that can be categorized in the categories of coercive, structure or autonomy support. Studies that include measures of feeding involving not only those based on coercive control, but also feeding regarding structure and autonomy support, can enrich our understanding of how caregivers are feeding their children and the effect it has on child weight. Using validated feeding measures that capture the diversity of food parenting practices that parents apply in LAC and how they relate to child weight status are needed. Learning about food parenting practices and styles in LAC can help identify food parenting practices that promote healthy weight in children in early and middle childhood with consideration to cultural diversity and context of LAC.

Although the objective of this review was focused on food parenting practices and feeding styles related to child weight status, we acknowledge food parenting practices and styles associations with child weight status requires to also include what and how much children eat as a necessary part of this association. Further research should also include this aspect to understand parent feeding as an influence in child weight status and approach relevant nutrition issues in countries of LAC.

## 5. Conclusions

The study systematically reviewed scientific evidence linking food parenting practices and feeding styles to weight status of early and middle childhood caregivers in LAC. Findings from all studies included in this review suggest associations between food parenting practices or feeding styles and child BMI, BMI-z scores or risk of overweight or obesity in Brazil, Chile and Mexico specifically. Therefore, the review partly met the objective to summarize the evidence of these relations in LAC because no countries from the Caribbean and several countries from Latin America have conducted studies examining this matter. Thus, the lack of research implies underrepresented countries of LAC, which narrows the findings to this review.

Existing studies have limitations, mainly related to convenience sample, small sample, heterogeneous feeding measures and weight related outcomes. Future research, including longitudinal studies, randomized samples, and feeding measures in the diverse nutritional contexts on LAC is warranted to assess the direction of associations and the long-term effect of feeding on weight status in early and middle childhood. Learning about food parenting practices and styles in LAC can help promote a healthy weight in children in childhood with consideration to cultural and contextual diversity in the region.

## Figures and Tables

**Figure 1 ijerph-19-02027-f001:**
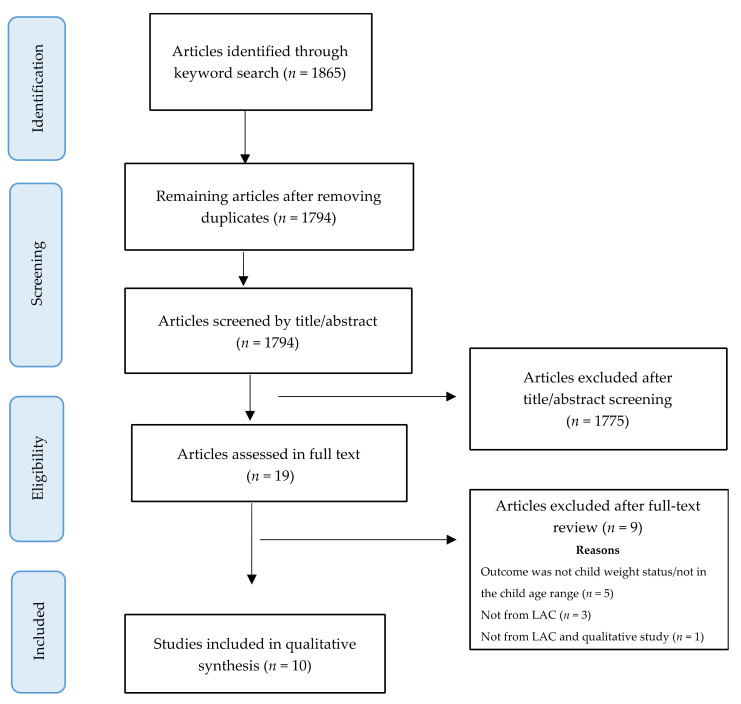
Study selection flowchart.

**Figure 2 ijerph-19-02027-f002:**
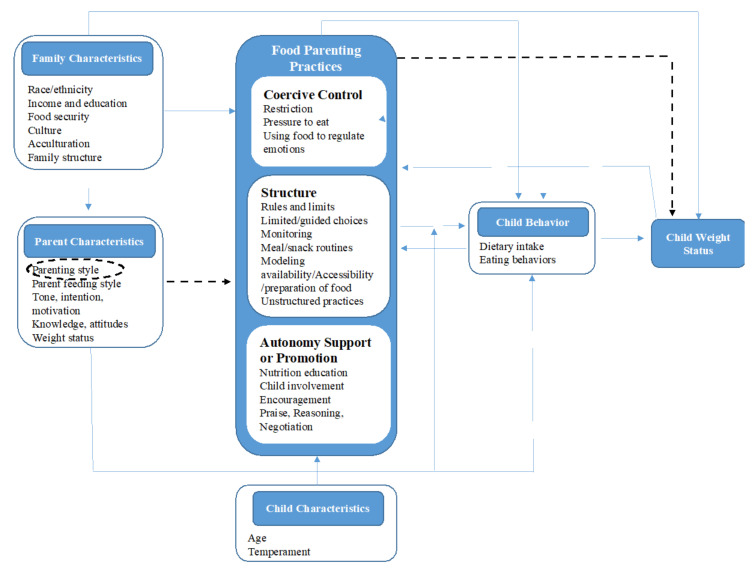
Adjusted content map of food parenting practices from Vaughn et al. [10]. The figure explains a conceptual map that hypothesizes mechanisms in which (1) family characteristics, (2) parent characteristics, and child characteristics relate to three higher-order categories of food parenting practices (Coercive Control, Structure, and Autonomy Support or Promotion) influencing child weight status. Food parenting practices directly relate to child weight status (dashed arrows) or indirectly through child behaviors such as dietary intake or eating behaviors. In this conceptual map, feeding styles indirectly relate to child weight status (dashed circle and dashed arrow) through food parenting practices or child behaviors. The map acknowledges how parents and family characteristics relate directly and indirectly to child weight status, and how child weight status, child behaviors, and child characteristics relate to the food parenting practices.

**Table 1 ijerph-19-02027-t001:** General description of studies included in the systematic review (*n* = 10).

Characteristics	N Studies
Total Number of Studies Selected	10
Publication dates	
2005–2010	3
2011–2015	2
2016–2020	5
Study Design	
Cohort/longitudinal	1
Cross-sectional	8
Case control	1
Study Age Groups	
2–5 years	4
2–8 years	2
3–11 years	1
6–12 years	3
Latin-American and Caribbean Countries	
Brazil	5
Chile	2
Mexico	3
Assessment of Food Parenting Practices and Feeding Styles (a questionnaire could be in more than one study)	
Child Feeding Questionnaire (CFQ)	4
Caregiver’s Feeding Style Questionnaire (CFSQ)	1
Parental Strategies for Eating and Activity Scale (PEAS)	1
Comprehensive Feeding Practices Questionnaire (CFPQ)	3
Toddler Feeding Questionnaire (TFQ)	1
Developed by authors	1

**Table 2 ijerph-19-02027-t002:** Summary of the main findings between food parenting practices/feeding styles and child weight outcomes.

Year	Country	Food Parenting Practices/Feeding Styles	Child Weight Outcomes	
			Child BMI	Likelihood of Being Obese/Overweight
Study 1 [33], 2019	Brazil	-Pressure to eat		BMI z-score related to lower likelihood of pressure to eat (OR = 0.86, *p* = 0.008)
Study 2 [41], 2018	Brazil	-Healthy eating guidance	(−) (β = −0.36, *p* < 0.05) in preescholers	
		-Pressure	(−) (β = −0.22, *p* < 0.05) in preschoolers	
		-Restriction for health	(+) (β = 0.09, *p* < 0.05) in school-age children	
		-Restriction for weight-control	(+) in preschool and school-aged children (β = 0.28, *p* < 0.001), with a larger effect among school-age children (β = 0.36, *p* < 0.001) compared to preschool aged children (β = 0.24, *p* < 0.05).	
Study 3 [35], 2014	Mexico	-Discipline	(+) (β = 0.101, *p* < 0.0026)	
		-Control (pressure the child to eat)	(+) (β = 0.332, *p* < 0.001)	
		-Limits	(+) (β = 0.095, *p* < 0.019)	
Study 4 [39] 2009	Chile	-Restriction	(+) (rho = 0.19 in boys, and rho = 0.27 in girls, *p* < 0.001)	Only in boys, restriction was related with higher likelihood of being obese (OR = 4.1, CI: 0.4–1.7, *p* = 0.001)
		-Pressure to eat	(−) (rho = −0.30 in boys, and rho = −0.36 in girls, *p* < 0.001)	In girls (OR = 0.2, CI: 0.06–0.4, *p* < 0.001) and boys (OR = 0.4, CI: 0.2–0.8, *p* < 0.001), high pressure to eat was associated with a lower likelihood of the child to be obese.
Study 5 [36], 2009	Chile	-Pressure to eat	(−) (r = −0.21, *p* < 0.05), on boys, in cross-sectional analyses.	
		-Restriction, monitoring and pressure to eat	Did not account for any variance in child BMI z-scores in retrospective analysis.	
Study 6 [34], 2017	Mexico	-Indulgent	higher BMI (mean = 1684)	
		-Uninvolved	higher BMI (mean = 1626)	
Study 7 [38], 2019	Brazil	-Restrictive	(+)	Likelihood of being overweight/obesity/severe obesity (OR = 2.18, *p* < 0.001)
Study 8 [32], 2011	Brazil	-Pressure to eat	(−)	Lower probability of weight excess (OR = 0.709, *p* = 0.010)
		-Restriction	(+)	Higher probability of weight excess (OR = 1.36, *p* < 0.001)
Study 9 [37], 2008	Brazil	-Restriction	(+)	Higher risk for child obesity (OR = 62.69; *p* = 0.0012)
Study 10 [40], 2019	Mexico	-Indulgent	(+) higher baseline BMI in boys and girls (β = 0.23; *p* = 0.02) at follow-up.	
		-Food use as a reward	(+) in girls, higher use of food as a reward at baseline positively associated with higher BMIz after 1 year follow-up (β = 0.31; *p* = 0.04).	
		-Restriction	(−) in girls, food restriction at baseline predicted lower BMIz at follow-up (β = −0.34; *p* = 0.03).	
		-Pressure to eat	No association with child BMI β = −0.033; *p* = 0.80)	

BMI: Body Mass Index; BMIz: z-scores of Body Mass Index; OR = Odds ratio.

**Table 3 ijerph-19-02027-t003:** Quality assessment of the studies based on STROBE and Lindsay et al. [23].

Authors (year)	(1)	(2)	(3)	(4)	(5)	(6)	(7)	(8)	(9)	(10)	Total
Costa et al. (2011) [32]	0	1	0	0	1	1	0	1	0	0	4
Flores-Peña et al. (2017) [34]	0	1	0	1	1	1	1	0	0	0	5
Freitas et al. (2019) [38]	0	1	0	1	0	1	1	1	0	1	6
Mulder et al. (2009) [36]	0	0	1	1	1	1	0	1	1	1	7
De Novaes et al. (2008) [37]	0	1	0	0	0	1	0	1	1	0	4
Santos et al. (2009) [39]	0	1	0	1	1	1	1	1	1	0	7
De Souza et al. (2019) [33]	0	1	0	1	1	1	1	1	1	1	8
Flores-Peña et al. (2014) [35]	0	1	1	1	0	1	0	1	0	0	5
Warkentin et al. (2018) [41]	0	1	0	1	1	1	1	0	0	1	6
Souto-Gallardo et al. (2019) [40]	1	1	0	1	1	1	0	0	0	1	6

Note: (1) Is the study longitudinal? (2) Does the paper describe the participants’ eligibility criteria? (3) Were study participants randomly selected (or representative of the study population)? (4) Did the paper report information about the measures (including validity in previous studies), including references used to assess food parenting practices? (5) Did the study include information about acceptable reliability of the instrument used to assess food parenting practices? (6) Did the paper report how child weight status was assessed in children participating in the study? (7) Did the study provide information about power calculation to detect hypothesized relationships? (8) Did the study report the number of individuals who completed each of the different measures? (9) Did the participants/respondents complete at least 80% of measures? (10) Did analyses account for confounding factors?

## Data Availability

Not applicable.

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
