# Peer review of "Food Parenting Practices and Feeding Styles and Their Relations with Weight Status in Children in Latin America and the Caribbean"

_ijerph, 2022, doi:10.3390/ijerph19042027_

Round 1
Reviewer 1 Report
Perez L. et. al. manuscript titled “Food Parenting Practices and Feeding Styles, and their Relations with Weight Status in Children in Latin America and the 3 Caribbean” shows the significant associations between food parenting practices and feeding styles and child weight status and their BMI. Although the review and the methods are extensive and study is promising, following point may improve manuscript.
Major points
- Table 2 feels redundant and can authors simple do as case 1 or study 1 and present studies.
- Can authors prepare brief table in with points with how there is weight linking to the food status. For example: for feeding style how the indulgent feeding was corresponding to high BMI. The figure is good representation, but table would be more helpful in associating each parameter to the outcome.
Author Response
Reviewer 1
Comments and Suggestions for Authors
Perez L. et. al. manuscript titled “Food Parenting Practices and Feeding Styles, and their Relations with Weight Status in Children in Latin America and the 3 Caribbean” shows the significant associations between food parenting practices and feeding styles and child weight status and their BMI. Although the review and the methods are extensive and study is promising, following point may improve manuscript.
Major points
1. Table 2 feels redundant and can authors simple do as case 1 or study 1 and present studies.
Response: Many thanks for this comment, we changed the redundant information in the Table 2, which we also moved as a supplementary table because another reviewer asked landscape format of it and the template of the journal in Word does not allow for the requested format. Please, see supplementary Table S1. Also, another reviewer asked for a simpler table, which we left in the article instead of the Table 2.
2. Can authors prepare brief table in with points with how there is weight linking to the food status. For example: for feeding style how the indulgent feeding was corresponding to high BMI. The figure is good representation, but table would be more helpful in associating each parameter to the outcome.
Response: We appreciate this suggestion to improve the understanding of these relationships. Please, see Table 2 in the revised article.
Reviewer 2 Report
The authors presented a well-written review work on food parenting practices feeding styles and connected them to understand child weight. More interestingly they focused on Latin America and the Caribbean. To date, no study was performed focusing on this region to highlight child weight issue. I thoroughly enjoyed reviewing the work. All sections were clearly explained and therefore the work is recommended to accept for publication.
However, there could be some very minor issues that need to be addressed, like-
L29- Sustainable Development - all in small letters.
L118- Correct spelling of feeding. Suggest making a careful spell-check of the whole manuscript.
L183- What does both authors mean? Who are they?
L200-201- Finally 10 papers came out only which satisfied the design of the exp. This number is really low to generalize an idea. Moreover, those conducted only in 3 countries and NONE in the Caribbean. Although this issue has been highlighted later in the ‘limitations’ section, I suggest to add some words/ a couple of sentences to justify this condition.
L223- The authors used used a case control design only which is a limiting factor of the manuscript. Please add some words to justify.
L239- Very short caption of the Fig. Please tell a bit more about what the fig is.
L382- child weight status and other weight-related outcomes. Please explain the reason/ mechanism of this outcome.
L459-… Brazil, Chile and Mexico specifically. These are already reported previously. How this is relevant to the manuscript. The authors need to connect that outcomes with their objective.
Author Response
Reviewer 2
Comments and Suggestions for Authors
The authors presented a well-written review work on food parenting practices feeding styles and connected them to understand child weight. More interestingly they focused on Latin America and the Caribbean. To date, no study was performed focusing on this region to highlight child weight issue. I thoroughly enjoyed reviewing the work. All sections were clearly explained and therefore the work is recommended to accept for publication.
However, there could be some very minor issues that need to be addressed, like-
L29- Sustainable Development - all in small letter
Response: Many thanks for this improvement. Please refer to the line 30 to check this change.
L118- Correct spelling of feeding. Suggest making a careful spell-check of the whole manuscript.
Response: We appreciate this suggestion, an English editor has edited the entire article.
L183- What does both authors mean? Who are they?
Response: Many thanks for this comment, we have added the initials of the authors to make this information more precise. Please, see line 185 and 209.
L200-201- Finally 10 papers came out only which satisfied the design of the exp. This number is really low to generalize an idea. Moreover, those conducted only in 3 countries and NONE in the Caribbean. Although this issue has been highlighted later in the ‘limitations’ section, I suggest to add some words/ a couple of sentences to justify this condition.
Response: We appreciate this suggestion, we expanded this explanation in lines 232-233 and in the limitations section (lines 485 to 488).
L223- The authors used used a case control design only which is a limiting factor of the manuscript. Please add some words to justify.
Response: Many thanks for suggesting this information, we included an explanation to justify inclusion of the article with this study design. Please, see lines 252-256.
L239- Very short caption of the Fig. Please tell a bit more about what the fig is.
Response: We appreciate this suggestion, we expanded the explanation of Figure 2. Please see lines 275 to 285.
L382- child weight status and other weight-related outcomes. Please explain the reason/ mechanism of this outcome.
Response: Many thanks for suggesting to add this information, we included an explanation about the mechanisms of the outcomes regarding the food parenting constructs. Please see lines 478 to 483.
L459-… Brazil, Chile and Mexico specifically. These are already reported previously. How this is relevant to the manuscript. The authors need to connect that outcomes with their objective.
Response: We appreciate this comment, we added information to indicate the connection between the findings in the three countries and the objectives of the review. Please see lines 485 to 488 in the Limitations section.
Reviewer 3 Report
The authors systemically reviewed papers to discuss associations between food parental feeding and feeding style and weight status of 2- to 12 years old children in Central and South America as well as Caribbean islands. The review is well written.
- Lines 34-36, are these problems common in these countries located in LAC? How many countries are in LAC? How many people live in LAC? Why is important to examine associations between food parenting practices, and feeding styles, and child’s weight status in the LAC region? No studies conducted in Caribbean islands were cited and discussed. Why the authors stated that studies carried out in LAC were reviewed?
- The authors did not mention ethnicity information of the subjects in LAC. Is it possible that ethnicity is one of reasons linked to the association of food parenting practices and feeding styles and weight status?
- Lin 21, what is the convenience sample?
- The age range of the children were 2-12 years old in the current study. Were significant differences found between 2-5 years old and 6-12 years old children regarding association between parental feeding and feeding style and weight status of children? The school aged children may have meals in school.
- Specific data are needed. For example, lines 307-308, “Among coercive feeding practices, restriction was associated with higher child BMI and higher risk for overweight/obesity in children in cross-sectional studies.” Can the detailed data be included to show the higher BMI and high risk for overweight/obesity in children?
- Landscape orientation is suggested for table 2.
- Line 186, 0-10?
- Line 194: two to 12 years old? 2 to 12 years old.
- Lines 200-202, 10 articles and 7 studies? Table 1, 10 studies?
- Line 281, Table 3?
Author Response
Reviewer 3
Comments and Suggestions for Authors
The authors systemically reviewed papers to discuss associations between food parental feeding and feeding style and weight status of 2- to 12 years old children in Central and South America as well as Caribbean islands. The review is well written.
- Lines 34-36, are these problems common in these countries located in LAC? How many countries are in LAC? How many people live in LAC?
Response: Many thanks for asking about the nutritional situation in LAC, and demographic information. We added this information in lines 66 and 67 and lines 43-54.
- Why is important to examine associations between food parenting practices, and feeding styles, and child’s weight status in the LAC region?
Response: The relevance of examining these associations relies on gaining understanding about the problem of malnutrition, especially due to the family factors associated with overweight in children, a trend that has been increasing in the region of LAC. Food parenting practices and feeding styles are influenced by their culture and others aspects of the context in which families live such as those mentioned in the introduction. We have expanded this information in lines 127 to 132.
3. No studies conducted in Caribbean islands were cited and discussed. Why the authors stated that studies carried out in LAC were reviewed?
Response: We could not find the lines you referred to, but we expanded the explanation about this point in the limitations section. Please, see lines 485 to 488 and conclusions sections lines 581-585.
4. The authors did not mention ethnicity information of the subjects in LAC. Is it possible that ethnicity is one of reasons linked to the association of food parenting practices and feeding styles and weight status?
Response: We appreciate this comment. The studies included in the review did not report ethnicity. We added the lack of this information as part of the limitations of the articles, please see lines 510 to 521.
5. Lin 21, what is the convenience sample?
Response: Many thanks for this comment, we revised this sentence and completed the information. Please, see line 21 and 22.
6. The age range of the children were 2-12 years old in the current study. Were significant differences found between 2-5 years old and 6-12 years old children regarding association between parental feeding and feeding style and weight status of children? The school aged children may have meals in school.
Response: Many thanks for these comments. Only some of the articles presented associations of interest according to age groups and this is why we did not previously present the findings by child age. We now added in the results section (lines 333 to 366) information of the associations separated by age group as was available in the studies.
7. Specific data are needed. For example, lines 307-308, “Among coercive feeding practices, restriction was associated with higher child BMI and higher risk for overweight/obesity in children in cross-sectional studies.” Can the detailed data be included to show the higher BMI and high risk for overweight/obesity in children?
Response: Many thanks for this suggestion, we added this information for clarification. Please, see lines 396 to 404.
8. Landscape orientation is suggested for table 2.
Response: Many thanks for this comment, we changed the orientation of the table, but this table is now included as a supplementary table because the template of the journal for Word does not allow a landscape format of the table. Please, see supplementary Table S1.
9. Line 186, 0-10?
Response: Many thanks for this comment, we modified it. Please, see line 209.
10. Line 194: two to 12 years old? 2 to 12 years old.
Response: Many thanks for this comment, we modified the word to a number. Please, see line 226.
11. Lines 200-202, 10 articles and 7 studies? Table 1, 10 studies? ()
Response: We would kindly ask for more explanation about this comment, so that we can properly address it. We cannot see to find a reference to 7 studies in the previously submitted version of the manuscript. Please advise.
12. Line 281, Table 3?
Response: We apologize about this, we modified this mistake. Please, see line 368.
Reviewer 4 Report
Summary
The authors assessed studies investigating children’s weight status, food parenting practices and feeding styles of early and middle childhood caregivers in Latin America and the Caribbean in a systematic review. Although the general topic and the comparison to findings in other parts of the world are interesting, there are some issues that need to be addressed before publication. I therefore recommend major revision of the manuscript.
Major Issues
- The three mentioned supplementary tables are not presented and could therefore not be reviewed.
- L 186-187: Please add an explanation how study quality scores are to be interpreted, e.g. which score is considered a well-conducted study in terms of quality? This is also missing when the authors report on the actual study quality of included studies (l. 281-286)
- Table 2 is somewhat difficult to comprehend since a lot of information is given in a text format as well as a lot of different information together in one column (e.g. study design, sample size, sample characteristics etc. are all given in one column). I think the table would benefit from having more columns for the respective characteristics to facilitate finding specific information.
- Please revise the paragraph on coercive feeding practices (l. 353-364) for clarity as it remains unclear for the reader what the conclusion from these findings are. Please also clarify what the relationship to “child weight status” means, are you referring to finding about overweight/obesity or underweight or something else?
Minor Issues
- L. 29-31 (“...to reach the goal of reducing both ends of child malnutrition and overweight and the undernutrition”): There seems to be an error, maybe the first “and” was meant to be another word?
- L. 159 “The two authors...” is inconclusive since there are more than two authors, maybe put “Two of the authors (initials of both authors)...”
- Can you add an indicator if interrater-reliability (e.g. cohen’s kappa) from before disagreements were resolved by discussion.
- L. 167: “presented” should be present tense
- Figure 1: There is an arrow missing between the second and third box from the top.
- Figure 1: Please add the reasons for excluding articles (as well as numbers of studies these applied to) to the flowchart.
- L. 313-316: Please check for grammar.
Author Response
Reviewer 4
Comments and Suggestions for Authors
Summary
The authors assessed studies investigating children’s weight status, food parenting practices and feeding styles of early and middle childhood caregivers in Latin America and the Caribbean in a systematic review. Although the general topic and the comparison to findings in other parts of the world are interesting, there are some issues that need to be addressed before publication. I therefore recommend major revision of the manuscript.
Major Issues
1. The three mentioned supplementary tables are not presented and could therefore not be reviewed.
Response: We apologize about the confusion. In the original submission, there were no supplementary tables. But, the revised submission includes 1 supplementary table, which was previously submitted as main tables.
2. L 186-187: Please add an explanation how study quality scores are to be interpreted, e.g. which score is considered a well-conducted study in terms of quality? This is also missing when the authors report on the actual study quality of included studies (l. 281-286).
Response: We appreciate this suggestion, we included an explanation to determine quality scores in the method section (lines 214 to 216). Also, we added information about study quality of in the results section (lines 368 to 369).
3. Table 2 is somewhat difficult to comprehend since a lot of information is given in a text format as well as a lot of different information together in one column (e.g. study design, sample size, sample characteristics etc. are all given in one column). I think the table would benefit from having more columns for the respective characteristics to facilitate finding specific information.
Response: We appreciate this suggestion, we made these changes to improve the table, which is now supplementary Table S1. Another reviewer asked that we change it to landscape format, but that is not possible in the Word template of the journal. Therefore, we include it now as a supplementary table and in landscape format.
4. Please revise the paragraph on coercive feeding practices (l. 353-364) for clarity as it remains unclear for the reader what the conclusion from these findings are. Please also clarify what the relationship to “child weight status” means, are you referring to finding about overweight/obesity or underweight or something else?
Response: Many thanks for this suggestion, we revised this paragraph in the conclusion section. We also included information to clarify the outcomes about child weigh status. Please refer to lines 447 to 456.
Minor Issues
L. 29-31 (“...to reach the goal of reducing both ends of child malnutrition and overweight and the undernutrition”): There seems to be an error, maybe the first “and” was meant to be another word?
Response: We appreciate this comment, we revised this sentence to improve understanding. Please, see lines 31 to 33.
L. 159 “The two authors...” is inconclusive since there are more than two authors, maybe put “Two of the authors (initials of both authors)...”
Response: We appreciate this comment, we added the authors' initials. Please, see line 185 and 209.
Can you add an indicator if interrater-reliability (e.g. cohen’s kappa) from before disagreements were resolved by discussion.
Response: We appreciate this comment, we added this information. Please see lines 210 and 211.
L. 167: “presented” should be present tense
Response: We appreciate this comment, we revised this word. Please see line 193 in the results section.
Figure 1: There is an arrow missing between the second and third box from the top.
Response: We appreciate this comment, we included the missing arrow in the figure 1.
Figure 1: Please add the reasons for excluding articles (as well as numbers of studies these applied to) to the flowchart.
Response: Many thanks for asking this request. We completed this information, please refer to the flowchart.
L. 313-316: Please check for grammar.
Response: We appreciate this suggestion, an English editor has edited the entire article.
Round 2
Reviewer 1 Report
The manuscript is promising and is acceptable in the present form.
Reviewer 3 Report
All my questions have been addressed.
Reviewer 4 Report
Thank you for the adaptions of the manuscript which seems ready for publication now. Just a small comment, the mentioned arrow in figure 1 (between second and third box from the top) is still missing.